# Tunable Switching between Slow and Fast Light in the Graphene Nanodisks (GND)–Quantum Dot (QD) Plasmonic Hybrid Systems

**DOI:** 10.3390/nano13050834

**Published:** 2023-02-23

**Authors:** Ghadah M. Almzargah, Mariam M. Tohari

**Affiliations:** Department of Physics, College of Science, King Khalid University, P.O. Box 9004, Abha 61421, Saudi Arabia

**Keywords:** graphene nanodisks-self-assembled quantum dots hybrid plasmonic systems, near-infrared radiation, electromagnetically induced transparency, switchable slow-fast light

## Abstract

Plasmonic nanocomposites demonstrate unique properties due to the plasmonic effects, especially those with graphene within their structures, thereby paving the way to various promising applications. In this paper, we investigate the linear properties of the graphene-nanodisks--quantum-dots hybrid plasmonic systems in the near-infrared region of the electromagnetic spectrum by numerically solving the linear susceptibility of the weak probe field at a steady state. Utilising the density matrix method under the weak probe field approximation, we derive the equations of motion for the density matrix elements using the dipole--dipole-interaction Hamiltonian under the rotating wave approximation, where the quantum dot is modelled as a three-level atomic system of Λ configuration interacting with two externally applied fields, a probe field, and a robust control field. We find that the linear response of our hybrid plasmonic system exhibits an electromagnetically induced transparency window and switching between absorption and amplification without population inversion in the vicinity of the resonance, which can be controlled by adjusting the parameters of the external fields and the system's setup. The probe field and the distance-adjustable major axis of the system must be aligned with the direction of the resonance energy of the hybrid system. Moreover, our plasmonic hybrid system offers tunable switching between slow and fast light near the resonance. Therefore, the linear properties obtained by the hybrid plasmonic system can be employed in applications such as communication, biosensing, plasmonic sensors, signal processing, optoelectronics, and photonic devices.

## 1. Introduction

Globally, extensive efforts has been dedicated to investigating the factors and mechanisms that optimise the optical properties of materials, which can lead to novel uses of the optical properties in various photonic applications [1]. Many applications can be described by linear optics, such as lenses, waveplates, mirrors, diffraction gratings, beam splitters, and phase shifters, which are commonly used applications in the field of linear optics.

The linear optical properties of many different kinds of materials have been discussed—for instance, those of solids, which is considered a prime topic in basic research and also for industrial applications such as optoelectronic devices [2]. Furthermore, linear optical properties have been extensively investigated in two-level and multi-level atomic systems. Fundamental optical properties have been theoretically and experimentally studied for these different atomic systems under various experimental conditions [3,4,5,6,7]. Indeed, the linear and nonlinear optical properties related to the phenomena of electromagnetically induced transparency (EIT) in the vicinity of the resonance can be controlled in multi-level atomic systems due to the atomic coherence induced in such systems [7,8,9].

Plasmonic nanostructures are important in nanophotonics due to the remarkable optical properties provided by these nanostructures [10]. Metallic nanoparticles illustrate distinctive physical phenomena called localised surface plasmons (LSPs) resulting from the collective oscillations of the free electrons on the surfaces of the metallic nanoparticles with resonance frequencies. The localised surface plasmon resonances provide characteristic properties for these metallic nanostructures, such as localising light in the nanoscale, making these structures excellent candidates for applications in the field of nanophotonics [11]. Moreover, the metallic nanoparticle surface plasmons can offer a strong local electromagnetic field that can enhance the light–matter interaction and optical properties of the system [12]. Modifying the surface structure of the metallic substance can manipulate the properties of the surface plasmons, which can be utilised in developing novel photonic devices and creating miniaturised photonic circuits whose lengths are relatively small—e.g., in data storage, subwavelength optics, and bio-photonics [10].

Plasmonic nanostructures offer geometric tunability to control the plasmon resonance and the optical properties of the plasmonic nanostructures [13,14,15,16]. Noble metals are mainly used as plasmonic materials due to the high density of charge carriers within their structures [17]. However, noble metals have some disadvantages that can limit their utility in optical processing devices; for instance, they are difficult to tune and display large Ohmic losses [18]. On the other hand, graphene can be considered an excellent plasmonic material with high confinement, a long propagation distance [19], and a low damping rate due to the high mobility of the graphene charge carriers [20]. Since the energy losses in plasmonic nanostructures are an essential defect in such structures, a gain medium such as quantum dots (QDs) can be inserted to compensate for energy loss within the hybrid system.

Moreover, the optical properties of the nanoplasmonic hybrid systems can be controlled and modified by manipulating these systems’ subwavelength structures, leading to characteristic properties and interesting phenomena such as superluminal propagation [21]. M.Tohari et al. have examined the linear optical properties in hybrid systems consisting of metal nanoparticles, graphene nanodisks, and quantum dots. It has been found that these linear properties can be tuned by the geometrical parameters of the system’s components and the interacting field’s parameters [20,21,22,23].

The near-infrared (near-IR) region of the electromagnetic spectrum has various applications in different areas, such as in medical diagnostics, therapies, and communications. Indeed, the near-IR region of radiation is distinguished by low attenuation and deep penetration with less damage due to the low energy of this region of the electromagnetic spectrum. Graphene nanostructures can create graphene surface plasmons (GSPs) with characteristic properties in the THz region, such as low damping rates and long propagation distances. However, for shorter wavelengths of the near-IR region of the spectrum, it is challenging to use the graphene plasmons because the chemical-doping methods can shift the Fermi level in graphene only for certain values [20]. Fortunately, practical techniques have been established to obtain graphene surface plasmons in the near-IR region, such as utilising the full scalable block copolymer self-assembly method [24,25]. Therefore, it is possible to control the chemical doping process in graphene, which can offer tunable graphene plasmonic devices in the region of near-IR [20,26].

In this paper, we numerically investigate the linear optical properties of graphene-nanodisks (GNDs)–quantum-dots (QDs) plasmonic hybrid systems in the near-IR region of the electromagnetic spectrum using the density matrix method under the rotating wave approximation to derive the equations of motion that describe the hybrid system’s dynamics at a steady state, where the self-assembled quantum dot is modelled as a three-level atomic system of Λ configuration interacting with a weak probe field and a strong control field.

## 2. Theoretical Formalism

We examined the light–matter interaction utilising a semiclassical model, where we considered a nanoplasmonic hybrid system consisting of GNDs and QDs embedded in a dielectric medium of indium phosphide (InP), as illustrated in Figure 1. The monolayer GNDs can provide strong confinement of the electromagnetic field beyond the diffraction limit as localised surface plasmons (LSPs) and offer remarkable properties due to the very high mobility of its charge carriers [19,27,28]. On the other hand, the gain medium is a self-assembled semiconductor quantum dot modelled as a three-level atomic system of Λ configuration, as shown in Figure 1.

The excitons within the InAs QD will be generated by the transitions |2〉↔|1〉 (|3〉↔|1〉), with transition frequency $\omega_{12}$ ($\omega_{13}$), and dipole moment $\mu_{12}$ along *z*-direction ($\mu_{13}$ along the *x*-direction), due to the interaction with the external probe field (control field). The probe field of Rabi frequency $\Omega_{p}=\frac{\mu_{12}E_{p}}{\hbar }$ will induce the transition between the ground state |2〉 and the excited state |1〉, and the control field of Rabi frequency $\Omega_{c}=\frac{\mu_{13}E_{c}}{\hbar }$ will cause the transition between the metastable state |3〉 and the excited state |1〉.

Note that the excited state will decay with the decay rates $\gamma_{12}$ and $\gamma_{13}$, and the metastable state |3〉 will decay with a decay rate $\gamma_{32}$ that is relatively small. Moreover, the two external fields interact with the GND, inducing localised surface plasmons (LSPs) on the interface between the GND and the surrounding dielectric medium (InP). The induced graphene surface plasmons (GSPs) with plasmonic frequency $\omega_{GSP}$ will produce a dipole electric field interacting with the InAs QD. Likewise, the excitons created due to the interaction with the external fields will generate a dipole electric field that interacts with the graphene surface plasmons (GSPs). Therefore, the interaction between the dipole electric fields induced in the GND and the QD will be through the dipole–dipole interaction (DDI) [19]. Indeed, this DDI is considered very strong due to the enhancement of the local dipole field in the GND, resulting in a substantial energy transfer between the GND and the InAs QD [19,27]. The induced DDI can be used to examine the coupling strength of the excitations between the hybrid system’s components.

In order to find the Hamiltonian that can describe the DDI between the GND and QD, we take the dipole electric field felt by the atomic system (QD), which can be written as follows:(1)$$E_{j}^{DDI}=\sum_{i,j}\frac{\hbar }{\mu_{1i}}\left[\Omega_{j}\left(\frac{1}{\epsilon_{dq}}+\Pi^{x,z}\right)+\Lambda_{i}^{x,z}\rho_{1i}\right]$$
where the index j≡p,c indicates the external fields, *p* refers to the probe field (polarised in *z*-direction), *c* refers to the control field (polarised in *x*-direction), and $\Omega_{j}$ is the Rabi frequency of the external fields, where $\Omega_{j}=\frac{E_{j}\mu_{1i}}{\hbar }$. $\rho_{1i}$ indicates the atomic coherence [23], and the index i≡2,3 refers to the atomic states in the QD, |2〉 and |3〉, which are coupled to the upper excited state |1〉 through $\rho_{1i}$. $\epsilon_{dq}=\left(2\epsilon_{d}+\epsilon_{q}\right)/3\epsilon_{d}$ is the effective dielectric constant [19,23], $\epsilon_{d}$ is the dielectric constant of the background dielectric medium, and $\epsilon_{q}$ is the dielectric constant of the QD. The system’s parameters $\Pi^{x,z}$ and $\Lambda_{i}^{x,z}$ are written as follows: (2)$$\Pi^{x,z}=\frac{g^{x,z}\alpha_{g}^{x,z}}{4\pi \epsilon_{dq}R_{QG}^{3}}$$
(3)$$\Lambda_{i}^{x,z}=\frac{g^{2(x,z)}\alpha_{g}^{x,z}\mu_{1i}^{2}}{{\left(4\pi \right)}^{2}\epsilon_{0}\epsilon_{dq}^{2}\hbar R_{QG}^{6}}$$
where $g^{x,z}$ is the polarisation direction. $\mu_{1i}$ are the dipole moments corresponding to the excitonic transitions |2〉↔|1〉 along *z*-direction and |3〉↔|1〉 along *x*-direction. $R_{QG}$ is the centre-to-centre distance between the QD and GND. The first two terms of Equation (Equation 1) describe the dipole field induced by the two external fields. The third term of Equation (Equation 1) is a polarisation term, since the external fields polarise the QD, and the QD, in turn, polarises the GND [23]. $\alpha_{g}^{x,z}$ is the shape-dependent polarisability of the GND in directions *x* and *z*; it can be written as [23]: (4)$$\alpha_{g}^{x,z}=\frac{4\pi V_{G}\left[\epsilon_{g}\left(\omega \right)-\epsilon_{d}\right]}{3\epsilon_{d}+3\zeta_{x,z}\left[\epsilon_{g}\left(\omega \right)-\epsilon_{d}\right]}$$
where $V_{G}$ is the GND volume. $\zeta_{x,z}$ is the depolarisability factor that can be given for a very flat and thin nanodisk of graphene, where $L_{x}=L_{y}>>L_{z}$ as $\zeta_{z}≃1-\frac{\pi }{2}\frac{L_{z}}{L_{x}}$ and $\zeta_{x}=\zeta_{y}≃\frac{\pi }{4}\frac{L_{z}}{L_{x}}$ [19], and the GND volume, in this case is $V_{G}=\left(\pi L_{x}^{2}\right)L_{z}$. Regarding the dielectric constant of graphene $\epsilon_{g}$, we considered the inter-band and intra-band transition contributions [23,29].

Therefore, the DDI Hamiltonian that corresponds to the dipole electric field $E_{j}^{DDI}$ can be written in the following form: (5)$$H_{DDI}=-\sum_{i,j}\hbar \left[\Omega_{j}\left(\frac{1}{\epsilon_{dq}}+\Pi^{x,z}\right)+\Lambda_{i}^{x,z}\rho_{1i}\right]\sigma_{1i}+H.C$$
For *z*-polarised probe field and *x*-polarised control field interacting with the plasmonic hybrid system depicted in Figure 1, the total Hamiltonian in the interaction picture [30] can be written for the three-level atomic system of Λ configuration under the rotating wave approximation (RWA) [31,32] as follows:(6)$$H_{\Lambda }=\hbar \left(\Delta_{p}\sigma_{11}+\Delta_{pc}\sigma_{33}\right)-\hbar \left[\Omega_{p}\left(\frac{1}{\epsilon_{dq}}+\Pi^{z}\right)+\Lambda_{2}^{z}\rho_{12}\right]\sigma_{12}-\hbar \left[\Omega_{c}\left(\frac{1}{\epsilon_{dq}}+\Pi^{x}\right)+\Lambda_{3}^{x}\rho_{13}\right]\sigma_{13}+H.C$$
The Hamiltonian is written in terms of the one photon detuning $\Delta_{p}$ and the two-photon detuning $\Delta_{pc}$, where $\Delta_{pc}=\Delta_{p}-\Delta_{c}$ and $\Delta_{p}=\omega_{p}-\omega_{12}$, where $\omega_{p}$ is the frequency of the probe field. $\Delta_{c}=\omega_{c}-\omega_{13}$ is the control field’s detuning, where $\omega_{c}$ is the frequency of the control field. Generally, $\Delta_{j}$, where (j≡p,c) refers to the detuning of the external fields. $\sigma_{11}$ and $\sigma_{33}$ are projection operators onto the upper and lower states. Finally, $\sigma_{1i}$ (where i≡2,3) are the flipping operators which connect the optical transitions within the system [23,30].

Utilising the density matrix method, where the time evolution of the density matrix elements can be studied by the Lindblad master equation (Equation [33,34]), we can derive the equations of motion for the density matrix elements $\rho_{ik}$ (where i,k≡1,2,3), which can be written as follows:(7)$$\frac{\partial \rho_{11}}{\partial t}=-\left(\gamma_{13}+\gamma_{12}\right)\rho_{11}+i\left[\Omega_{p}\left(\frac{1}{\epsilon_{dq}}+\Pi^{z}\right)+\Lambda_{2}^{z}\rho_{12}\right]\rho_{21}+i\left[\Omega_{c}\left(\frac{1}{\epsilon_{dq}}+\Pi^{x}\right)+\Lambda_{3}^{x}\rho_{13}\right]\rho_{31}+C.C$$
(8)$$\frac{\partial \rho_{22}}{\partial t}=\left(\gamma_{12}\rho_{11}+\gamma_{32}\rho_{33}\right)-i\left[\Omega_{p}\left(\frac{1}{\epsilon_{dq}}+\Pi^{z}\right)+\Lambda_{2}^{z}\rho_{12}\right]\rho_{21}+C.C$$
(9)$$\frac{\partial \rho_{33}}{\partial t}=-\gamma_{32}\rho_{33}+\gamma_{13}\rho_{11}-i\left[\Omega_{c}\left(\frac{1}{\epsilon_{dq}}+\Pi^{x}\right)+\Lambda_{3}^{x}\rho_{13}\right]\rho_{31}+C.C$$
(10)$$\frac{\partial \rho_{13}}{\partial t}=\left[\frac{-\gamma_{13}}{2}-\frac{\gamma_{12}}{2}-\frac{\gamma_{32}}{2}-i\Delta_{c}+i\Lambda_{3}^{x}\left(\rho_{33}-\rho_{11}\right)\right]\rho_{13}+i\left[\Omega_{p}\left(\frac{1}{\epsilon_{dq}}+\Pi^{z}\right)+\Lambda_{2}^{z}\rho_{12}\right]\rho_{23}+i\Omega_{c}\left(\frac{1}{\epsilon_{dq}}+\Pi^{x}\right)\left(\rho_{33}-\rho_{11}\right)$$
(11)$$\frac{\partial \rho_{12}}{\partial t}=\left[\frac{-\gamma_{13}}{2}-\frac{\gamma_{12}}{2}-i\Delta_{p}+i\Lambda_{2}^{z}\left(\rho_{22}-\rho_{11}\right)\right]\rho_{12}+i\Omega_{p}\left(\frac{1}{\epsilon_{dq}}+\Pi^{z}\right)\left(\rho_{22}-\rho_{11}\right)+i\left[\Omega_{c}\left(\frac{1}{\epsilon_{dq}}+\Pi^{x}\right)+\Lambda_{3}^{x}\rho_{13}\right]\rho_{32}$$
(12)$$\frac{\partial \rho_{32}}{\partial t}=\left[\frac{-\gamma_{23}}{2}-i\Delta_{pc}\right]\rho_{32}+i\left[\Omega_{c}^{*}\left(\frac{1}{\epsilon_{dq}^{*}}+\Pi^{*x}\right)+\Lambda_{3}^{*x}\rho_{31}\right]\rho_{12}-i\left[\Omega_{p}\left(\frac{1}{\epsilon_{dq}}+\Pi^{z}\right)+\Lambda_{2}^{z}\rho_{12}\right]\rho_{31}$$
Based on the above equations of motion, it can be seen that since $\Lambda_{i}^{x,z}$ is the dipole term, the real part of $\Lambda_{i}^{x,z}\left(\rho_{ii}-\rho_{11}\right)$ describes the non-radioactive energy shift for the system, and the imaginary part stands for the decay rate, which indicates enhancement for the QD spontaneous emission due to the presence of the plasmonic component (GND) by $\Lambda_{i}^{x,z}$. Therefore, controlling the spontaneous emission is possible through the structural and geometrical parameters of the hybrid system. Additionally, the strength of the interaction between the plasmons in GND and the excitons in the QD, represented by Rabi frequency $\Omega_{j}$, is boosted by the factor $\left(\frac{1}{\epsilon_{dq}}+\Pi^{x,z}\right)$ due to the DDI, which gives a strong coupling between excitons and the graphene surface plasmons for $\left(\frac{1}{\epsilon_{dq}}+\Pi^{x,z}\right)>1$, resulting in strong energy transfer and enhanced optical properties for the hybrid system.

In order to investigate the linear optical response of the plasmonic hybrid system, including absorption, dispersion, and group velocity $\nu_{g}$, we derive the linear susceptibility of the system. The equation that describes the linear susceptibility can be written as follows [30,35]: (13)$$\chi =\frac{N\mu_{12}^{2}}{\epsilon_{0}\hbar \Omega_{p}}\rho_{12}$$
By calculating $\rho_{12}$, we can obtain the formula for the linear susceptibility χ. To find $\rho_{12}$, we solve the equation of motion (Equation (Equation 11)) for $\rho_{12}$ at a steady state where $\frac{\partial \rho_{ik}}{\partial t}\equiv \rho_{ik}^{\prime }\approx 0$ (i,k≡1,2,3), and under the weak probe field approximation to eventually get: (14)$$\rho_{12}=\frac{i\Omega_{p}\left(\frac{1}{\epsilon_{dq}}+\Pi^{z}\right)}{\frac{\gamma_{13}}{2}+\frac{\gamma_{12}}{2}+i\Delta_{p}-i\Lambda_{2}^{z}+\frac{{\left|\Omega_{c}\left(\frac{1}{\epsilon_{dq}}+\Pi^{x}\right)\right|}^{2}}{\frac{\gamma_{23}}{2}+i\Delta_{pc}}}$$Therefore, the linear susceptibility can be written in the following form: (15)$$\chi \approx \frac{2iN\mu_{12}^{2}\left(\frac{1}{\epsilon_{dq}}+\Pi^{z}\right)}{\epsilon_{0}\hbar F}$$
where: (16)$$F=\frac{\gamma_{13}}{2}+\frac{\gamma_{12}}{2}+i\Delta_{p}-i\Lambda_{2}^{z}+\frac{\Omega_{c}^{2}{\left|\frac{1}{\epsilon_{dq}}+\Pi^{x}\right|}^{2}}{\frac{\gamma_{23}}{2}+i\Delta_{pc}}$$

The susceptibility can be defined as a frequency-dependent quantity χ=χ(ω) in a dispersive medium with frequency-dependent dispersion. Additionally, it is known that the envelope of an optical pulse that propagates in a medium has a velocity called the group velocity $\nu_{g}$; this group velocity can be written as: (17)$$\nu_{g}=\frac{c}{1+\frac{1}{2}Re\left[\chi \left(\omega_{p}\right)\right]+\frac{\omega_{p}}{2}\frac{\partial Re[\chi \left(\omega_{p}\right)]}{\partial \omega_{p}}}=\frac{c}{n_{g}}$$
where $n_{g}$ represents the group index. The group index can be defined as $n_{g}=\left(c/\nu_{g}\right)-1$ [9] for differentiating the propagation of subluminal fields from superluminal fields within the system. The situation when $n_{g}>0$ indicates the so-called "slow light" and subluminal field propagation within the system, whereas when $n_{g}<0$, it corresponds to the case of "fast light" and superluminal field propagation [9]. From (Equation (Equation 17)), it can be observed that fast light can be achieved when the derivative $\frac{\partial Re[\chi ]}{\partial \omega }$ is negatively large in the vicinity of resonance, whereas slow light can be obtained when the derivative $\frac{\partial Re[\chi ]}{\partial \omega }$ is positive and large [36].

## 3. Results and Discussion

To investigate the linear response of the GND-QD plasmonic hybrid system, we assumed a monolayer GND with a thickness of approximately one atom ($L_{z}=0.35$ nm) [19] and a radius of ($L_{y}=L_{x}=7$ nm), a room temperature of 300 K, and mobility of the charge carriers $\mu =10^{4}$ cm^2^/V.s. In addition, the graphene was doped at a Fermi energy of $E_{F}=0.51$ eV and embedded in a background medium of indium phosphide (InP) with a dielectric constant of $\epsilon_{d}=10$. For the above parameters, the extinction cross-section spectrum of the monolayer GND exhibits two plasmonic resonances due to the *x* and *z*-polarised fields, i.e., $E^{x}=\hbar \omega_{GSP}^{x}=0.18$ eV and $E^{z}=\hbar \omega_{GSP}^{z}=0.8$ eV, as seen in Figure 2. Since we are interested in the near-IR region, we chose the resonance in *z*-direction: $E^{z}=\hbar \omega_{GSP}^{z}=0.8$ eV.

In order to induce the energy transfer between the components of the system, the QD was chosen to be the (InAs/InP) self-assembled quantum dot with excitation energy 0.8 eV (∼1.55 μm) [37] in the near-IR region. This excitation energy matches that of the GND along z-direction. The InAs QD had an atomic density of $N=3\times 10^{21}$ m^−3^ [38] and a dielectric constant of $\epsilon_{q}=12$ [39]. The other parameters of the QD were set as follows: $\mu_{12}=\mu_{13}=1\times 10^{-28}C.m$ [38,40], with decay rates of $\gamma_{12}=\gamma_{13}=10^{9}\;s^{-1}$ [38] and $\gamma_{32}=0.35\gamma_{12}$ [20], where $\gamma_{12},\gamma_{13}>>\gamma_{32}$, and $\Omega_{c}\geq \gamma_{12}$ and $|\Omega_{c}{|}^{2}>>\gamma_{12}\gamma_{32}$ [35,41] to match the EIT conditions, with $\Omega_{p}$ to be $\Omega_{p}=0.01\Omega_{c}$ [20]. The centre-to-centre distance between the system’s components is $R_{QG}=L_{z}+R_{e}+2.5$ nm, where $R_{e}$ is the edge-to-edge distance and 2.5 nm is the typical radius for a QD [42,43]. Note that the polarised probe field and $R_{QG}$ are along the same direction as the hybrid system’s resonance energy (*z*-direction).

To examine the optical properties of the system, we plotted the linear susceptibility of the hybrid system versus the probe field’s frequency, as seen in Figure 3. We notice that the interaction of the hybrid system with the two electromagnetic fields under the above conditions can lead to the splitting of the absorption spectrum into two peaks and a minimum of absorption at the resonance $\omega_{p}/\omega_{12}=1$ (Figure 3b), which is associated with the anomalous dispersion near the resonance (Figure 3a). This phenomenon is called the electromagnetically induced transparency (EIT). It is known that the electromagnetically induced coherence of the electronic states in nanostructures results from quantum interference between the atomic states, and it might eliminate the atomic transition’s absorption and refraction at the resonance [9]. Additionally, we can control the system’s absorption and dispersion by adjusting the distance $R_{QG}$ between the GND and the QD. In (Figure 3), it can be seen that the linear susceptibility is enhanced for small distances $R_{QG}$, since it is known that the DDI inversely depends on the distance $R_{QG}$.

Generally, in three-level atomic systems, the linear optical response can be modified due to the interaction with a strong resonant field that is able to dress the atomic states, preventing the absorption of another weak resonant field between two paths of absorption, which result from the formation of two dressed states rather than one excited state |1〉 [44,45]. Furthermore, it is evident that for small Rabi frequencies of the control field $\Omega_{c}$, the susceptibility will be reduced to that of two-level systems, which appears in the absorption spectrum as a single peak of absorption at the resonance ($\omega_{p}/\omega_{12}=1$), as seen in Figure 4a—the dot-dashed line. Moreover, applying a stronger control field to the hybrid system will cause two peaks of absorption to emerge gradually until an EIT window appears at $\Omega_{c}=8\mathrm{THz}$ (Figure 4a, the solid line). Interestingly, there is a switch between negative and positive absorption in the vicinity of resonance at Rabi frequencies $\Omega_{c}$ between 9 THz and roughly 29 THz, where the range of switching increases as the Rabi frequency of the control field increases (Figure 4b). Furthermore, negative absorption arises precisely at resonance, just ahead of the emergence of the EIT window at Rabi frequencies ($\Omega_{c}$ = 5–7 THz) (Figure 4b).

Remarkably, a wider EIT window can be achieved by applying a stronger control field ($\Omega_{c}\geq 30$ THz) to the hybrid system (Figure 4c). It is evident from the equations of motion that the increments in $\Omega_{j}$, which represents the interaction strength, provide a solid coupling between the excitons of the QD and the graphene’s surface plasmons, resulting in optimisation of the system’s optical properties. Therefore, the system’s optical properties can be controlled by adjusting the interaction strength, $\Omega_{j}$, and the distance between the hybrid system’s components, $R_{QG}$. Generally, we can see that the system’s absorption (dispersion) fluctuates between positive and negative signs as the Rabi frequency of the control field changes until these fluctuations vanish for large values of $\Omega_{c}$ (Figure 5a,b). In (Figure 5c,d), it can be observed that the linear susceptibility χ does not change noticeably for modest shifts of the control field away from the resonance. As expected, the system’s absorption and dispersion decline as $\Delta_{c}$ increases (Figure 5c,d).

It is worth mentioning that fibre technology can provide continuous-wave (CW) lasers with large optical intensities. Interestingly, high-power CW fibre lasers can offer a laser power range of 1 kW to more than 100 kW. In 2013, E. A. Shcherbakov et al. generated the most powerful CW fibre laser system for industrial applications, with a power level of 101.3 kW and emitting at a wavelength range of 1070 nm [46]. In our numerical investigations, the Rabi frequency of the control field was at the THz level with an optical intensity of approximately 10^9^ W/m^2^, which could be obtained using the high-powered CW fibre lasers.

To determine whether the negative absorption (amplification) of the weak field and the switching between positive and negative absorption are due to a population inversion from the ground state |2〉 to the excited state |1〉, we plotted the population inversion of the system between the ground and excited states ($\rho_{11}-\rho_{22}$) versus the frequency of the weak probe field (Figure 6). Noticeably, the population is captured in the ground state |2〉, since the negative values of $\rho_{11}-\rho_{22}$ shown in Figure 6 indicate that the majority of the system’s population is in the ground state. Therefore, the switching between positive and negative absorption and the amplification at the resonance occur without population inversion due to the plasmonic effects of the GND within the hybrid system.

Figure 6a shows that even though most of the population is in the ground state, the probability of the population moving slightly to the excited state increases for smaller distances $R_{QG}$ and large values of $\Omega_{c}$ (Figure 6b). Furthermore, as the detuning of the control field increases, the system’s population in the excited state slightly reduces (Figure 6c).

Theoretically, it can be predicted that in a three-level atomic system in Λ configuration, the effect of ultrashort pulse amplification could occur without either population inversion or coherent optical pumping. This effect is due to the capturing of the population at the two lower states in a low-frequency excitation. Furthermore, this effect can be used to achieve amplification or generation at transitions where it is difficult for the population inversion to happen [47].

Thus, the plasmonic hybrid system provides a tunable EIT window, which is needed for many applications, such as signal transportation and quantum information. Moreover, the controllable switching between absorption and amplification near the resonance without population inversion has various potential applications, for instance, optical amplifiers, biosensors, and many photonic devices.

In addition, we examined the group index $n_{g}$ in the plasmonic hybrid system in a narrow range of frequencies around the resonance ($\omega_{p}/\omega_{12}=1$). Figure 7 shows the group index $n_{g}$ of the weak probe field propagating in the vicinity of the resonance of the QD impacted by the plasmonic effects of the GND.

In Figure 7, we notice that there is switching between $n_{g}>0$, which indicates slow light and subluminal propagation through the system, and $n_{g}<0$, which results in superluminal propagation via the system and fast light. In addition, the group index switching between fast and slow light is enhanced for small distances between the system’s components (Figure 7a). Further, the interaction of the system with a stronger control field provides a wider range of frequencies around the resonance for the switching between fast and slow light, as illustrated in (Figure 7b).

Hybrid systems of semiconductor quantum dots (SQDs) and metal nanoparticles (MNPs) have attracted incredible interest for their optical properties. These optical properties can be controlled by modifying the sizes and shapes of the system’s components [48]. Moreover, a tunable transparency window and switching of fast to slow light or vice versa have been investigated in the SQDs-MNPs systems [40,49]. Nevertheless, such hybrid systems do not exhibit switching between absorption and amplification, nor does the sign of the group index change rapidly near the resonance. Moreover, it has been found that the presence of the GND within the MNP-GND-QD hybrid system proposed by M.Tohari et al. can support the tunable switching between the positive and negative absorption near the resonance [21]. However, the switching between absorption and amplification is not rapid near the resonance. Remarkably, the GND-QD plasmonic hybrid system demonstrates quick, tunable switching between absorption and amplification without population inversion at a specific range of the control field’s Rabi frequencies. Then, for large values of $\Omega_{c}$, a tunable EIT can be observed, where there is still switching between absorption and amplification, but it is weaker for large values of the control field’s Rabi frequency. Additionally, the GND-QD plasmonic hybrid system achieves rapid switching between slow and fast light in the vicinity of the resonance. Indeed, the rapid switching between subluminal and superluminal propagation of the probe field through the system in the vicinity of the resonance is due to the presence of the GND’s plasmonic effects causing the switching between absorption and amplification without population inversion just ahead of the EIT window’s emergence, which in turn is associated with anomalous dispersion and fast light.

Interestingly, we found that the centre-to-centre distance between the system’s components must be in the direction of the polarised probe field, which was chosen to be along the direction of the excitation energy of the graphene surface plasmons (*z*-direction), which matches the excitation energy of the QD. Otherwise, the response of the plasmonic hybrid system to changing the distance $R_{QG}$ will deviate from what is known according to the DDI. The strength of this interaction should decrease significantly when the distance $R_{QG}$ increases. Indeed, when the distance $R_{QG}$ is perpendicular to the hybrid system’s resonance energy direction (the probe field’s direction) (Figure 8), we note that the hybrid system responds to adjusting the distance $R_{QG}$ in an opposite manner to what is expected. The optical properties exhibit higher values for larger distances between the system’s components (Figure 9) for a concise range of values before the hybrid system losses its sensitivity toward $R_{QG}$ (Figure 10). The reason behind this odd optical response of the hybrid system is that the interaction is in the direction of the polarised robust control field and not along the resonance direction of the hybrid system. Compared with the above results obtained by the setup shown in Figure 1, for the plasmonic hybrid system to respond to changing of the geometrical parameter $R_{QG}$, the distance between the system’s components $R_{QG}$ must be in the direction of the hybrid system’s resonance energy, which means the angle θ must be zero, and ϕ must be π/2 (as can be seen in Figure 1). Furthermore, if we return to the Equation (Equation 15), we can see that the numerator contains the factor $\Pi^{z}$. $\Pi^{z}$ will be negative when the distance $R_{QG}$ and the polarised probe field are perpendicular (θ=π/2). The odd direct relationship between the distance and the response of the optical properties of the hybrid system is explained by the negative sign of $\Pi^{z}$ with $R_{QG}^{3}$ in the denominator. Moreover, the terms ($i\Lambda_{2}^{z}$) and ($\left(\Omega_{c}^{2}{\left|\frac{1}{\epsilon_{dq}}+\Pi^{x}\right|}^{2}\right)/\left(\frac{-\gamma_{23}}{2}-i\Delta_{pc}\right)$) are included in the denominator of the equation of the system’s susceptibility (Equation (Equation 15)), and the factors $\Lambda_{2}^{z}$ and $\Pi^{x}$ inversely depend on $R_{QG}^{6}$ and $R_{QG}^{3}$, respectively; hence, $R_{QG}$ plays a significant role in all the optical properties of the system, and choosing $R_{QG}$ in the direction of the system’s resonance energy is critical in the GND-QD hybrid system.

Ultimately, a controllable EIT window and switching between absorption and amplification without population inversion in the vicinity of the resonance have been achieved in the region of near-IR due to the plasmonic effects generated by a single plasmonic component (GND) located near a QD. Additionally, tunable switching between slow and fast light in the near-IR region has been examined, which is useful in applications that need an optical switch between slow and fast light. This hybrid plasmonic system of GND and QD could be utilised in various applications in communications and infrared imaging, and in many photonic devices, such as photodetectors [50], plasmonic amplifiers, optical modulators [50], and plasmonic sensors.

## 4. Conclusions

We have examined the linear optical properties of the GND-QD plasmonic hybrid system in the near-IR region of the electromagnetic spectrum, where the self-assembled quantum dot (InAs/InP) was modelled as a three-level atomic system of Λ-type interacting with two external fields, the weak probe field and the strong control field. A numerical solution for the probe field’s linear susceptibility has been obtained using the density matrix method under the weak probe field approximation. In addition, we calculated the density matrix elements’ equations of motion, which describe the system’s dynamics, by utilising the DDI Hamiltonian at a steady-state and under the RWA. We have found that the linear optical response of our hybrid system exhibits an EIT window and switching between absorption and amplification without population inversion near the resonance, which can be tuned by controlling the parameters of the externally applied fields ($\Omega_{j}$ and $\Delta_{j}$) and the geometrical parameter of the hybrid system ($R_{QG}$), which must be in the same direction as the hybrid system’s resonance energy. In addition, our hybrid system offers tunable switching in the near-IR region between slow light with subluminal propagation and fast light associated with superluminal propagation of the probe field through the system. Interestingly, the setup alignment critically impacts the optical response of the plasmonic hybrid systems composed of graphene nanodisks and self-assembled quantum dots, which might promote the development of more efficient optoelectronic devices.

We hope this work will theoretically and experimentally encourage more investigations into the optical linear and nonlinear properties of the GND-QD plasmonic hybrid nanostructure in other regions of the electromagnetic spectrum to improve our understanding of such hybrid systems and their potential applications. 

## Figures and Tables

**Figure 1 nanomaterials-13-00834-f001:**
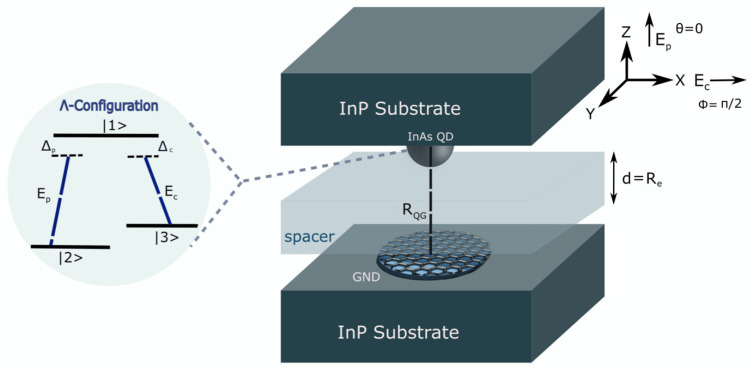
The proposed setup of the graphene nanodisk (GND)–quantum dot (QD) plasmonic hybrid system. In this setup, we use a spacer layer for controlling the centre-to-centre distance $R_{QG}$ between the GND and the self-assembled (InAs/InP) QD. The thickness of the spacer (d) can be considered as the edge-to-edge distance $R_{e}$.

**Figure 2 nanomaterials-13-00834-f002:**
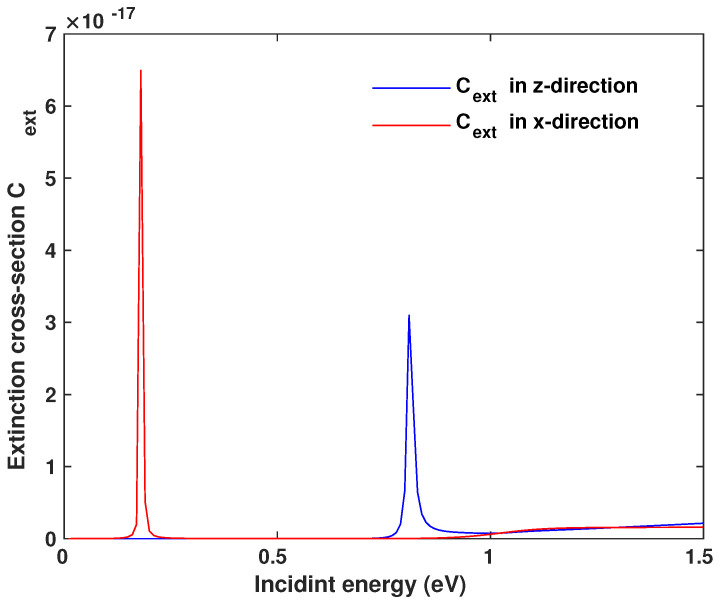
Extinction cross-section of a monolayer graphene nanodisk, where the two peaks indicate the resonances depending on the direction of the incident field.

**Figure 3 nanomaterials-13-00834-f003:**
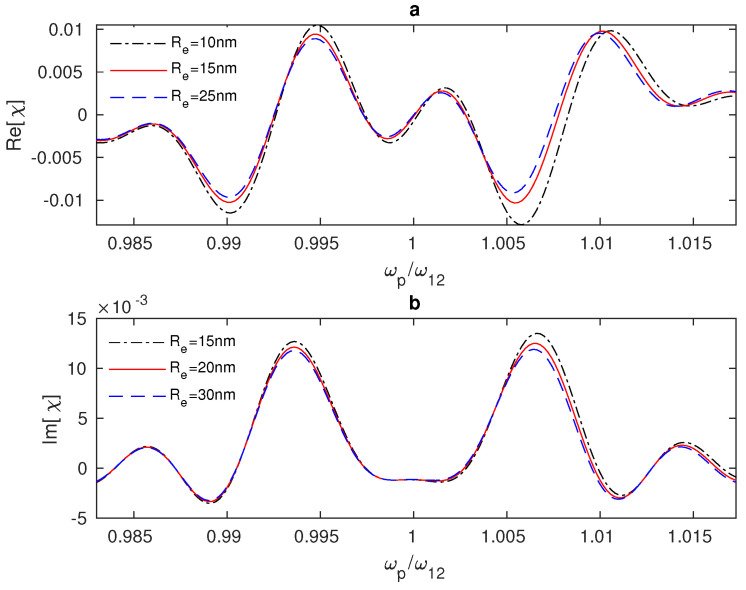
Linear susceptibility of the GND-QD plasmonic hybrid system versus the probe field frequency $\omega_{p}$. The hybrid system interacts with a resonant control field $\Delta_{c}=0$ at different edge-to-edge distances $R_{e}$ between the system’s components, where the Rabi frequency of the control field is $\Omega_{c}$ = 10 THz in (**a**) and $\Omega_{c}$ = 8 THz in (**b**).

**Figure 4 nanomaterials-13-00834-f004:**
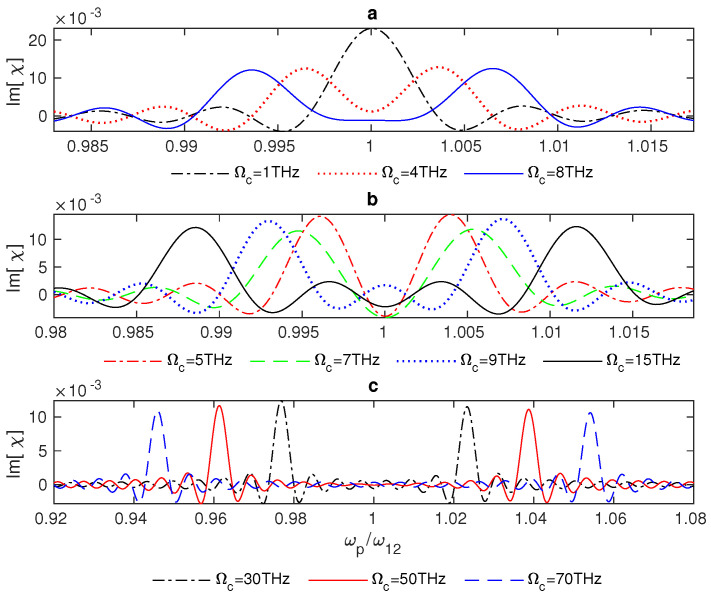
Absorption of the GND-QD plasmonic hybrid system as a function of the probe field frequency $\omega_{p}$. The system interacts with a resonant control field $\Delta_{c}=0$ at low (**a**), moderate (**b**), and high (**c**) values of Rabi frequency of the control field, and the edge-to-edge distance is $R_{e}$ = 20 nm.

**Figure 5 nanomaterials-13-00834-f005:**
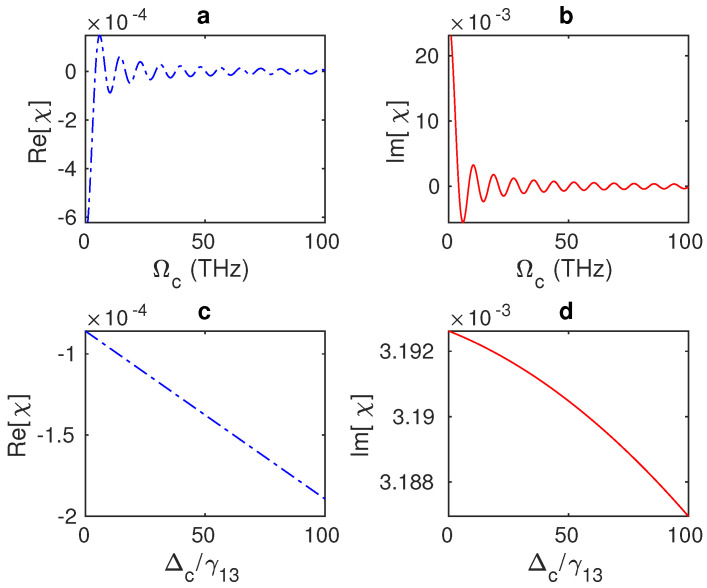
(**a**,**b**) Absorption and dispersion spectra of the GND-QD hybrid system at a resonant probe field ($\Delta_{p}=0$) as a function of the Rabi frequency of the control field $\Omega_{c}$, where the edge-to-edge distance is $R_{e}$ = 20 nm and $\Delta_{c}$ = 0. (**c**,**d**) The system’s absorption and dispersion versus the detuning of the control field $\Delta_{c}$ at an edge-to-edge distance of $R_{e}$ = 20 nm and $\Omega_{c}$ = 10 THz.

**Figure 6 nanomaterials-13-00834-f006:**
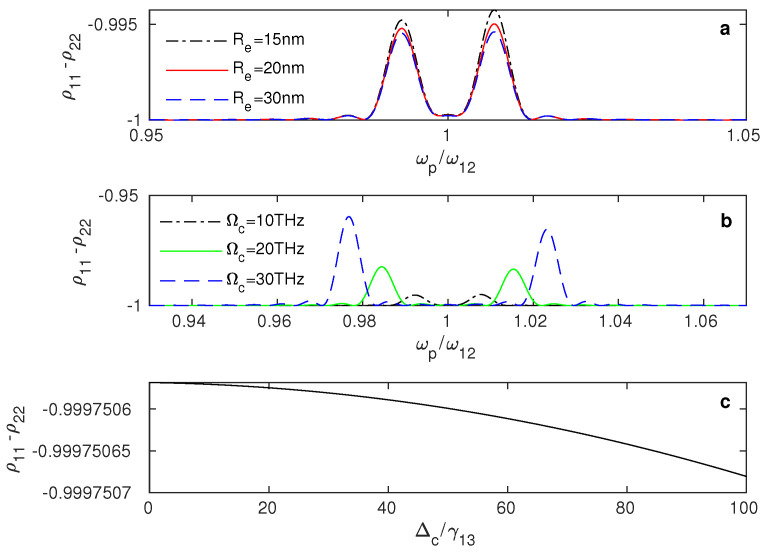
Population of the plasmonic hybrid system between the ground and excited states versus the probe field frequency $\omega_{p}$ at different values of the edge-to-edge distance $R_{e}$ (**a**), where Rabi frequency of the control field is 10 THz and the detuning of the control field is 0. (**b**) The population at different values of the control field’s Rabi frequency, $\Omega_{c}$, with an edge-to-edge distance of 20 nm and detuning of the control field of 0. (**c**) The system’s population as a function of the control field’s detuning $\Delta_{c}$, where the edge-to-edge distance is $R_{e}$ = 20 nm, and Rabi frequency is $\Omega_{c}$ = 10 THz.

**Figure 7 nanomaterials-13-00834-f007:**
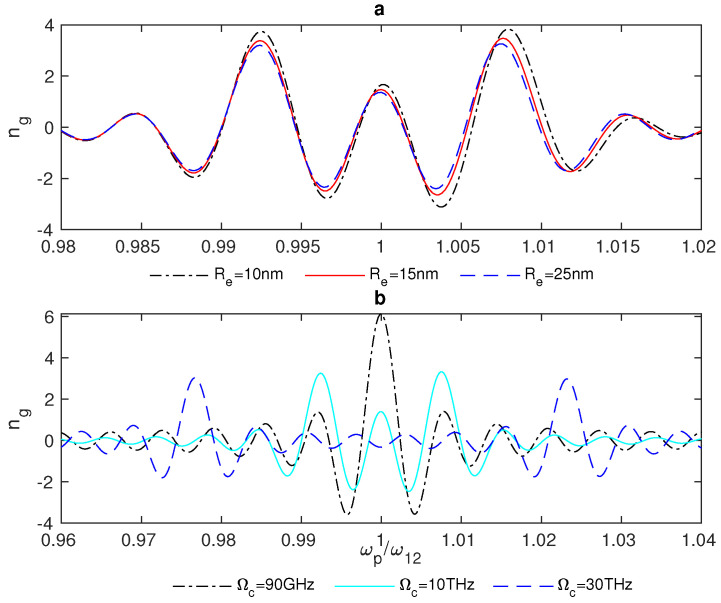
Group index $n_{g}$ in the GND-QD plasmonic hybrid system at different edge-to-edge distances between the system’s components $R_{e}$ (**a**), where $\Omega_{c}$= 10 THz and the system interacts with a resonant control field with $\Delta_{c}$ = 0. (**b**) The system’s group index at different values of the Rabi frequency of the control field, $\Omega_{c}$, where the edge-to-edge distance $R_{e}$ = 20 nm and $\Delta_{c}$ = 0.

**Figure 8 nanomaterials-13-00834-f008:**
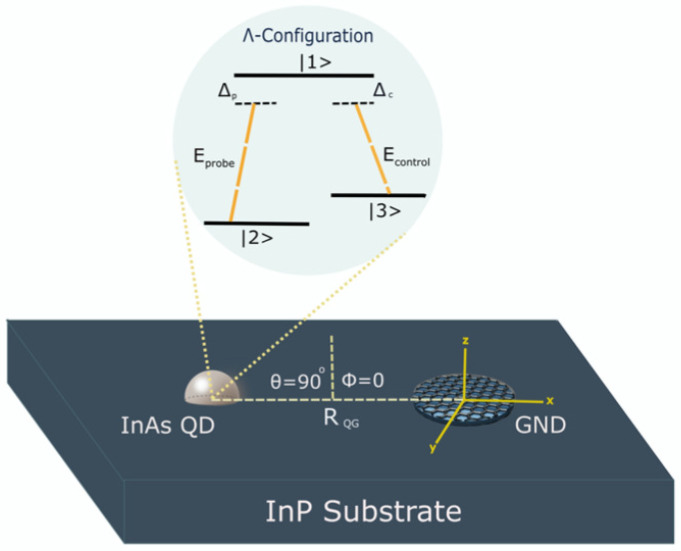
The setup of the graphene nanodisk (GND)-quantum dot (QD) plasmonic hybrid system when the distances $R_{QG}$ between the system’s components are perpendicular to the direction of the hybrid system’s resonance energy (z-direction).

**Figure 9 nanomaterials-13-00834-f009:**
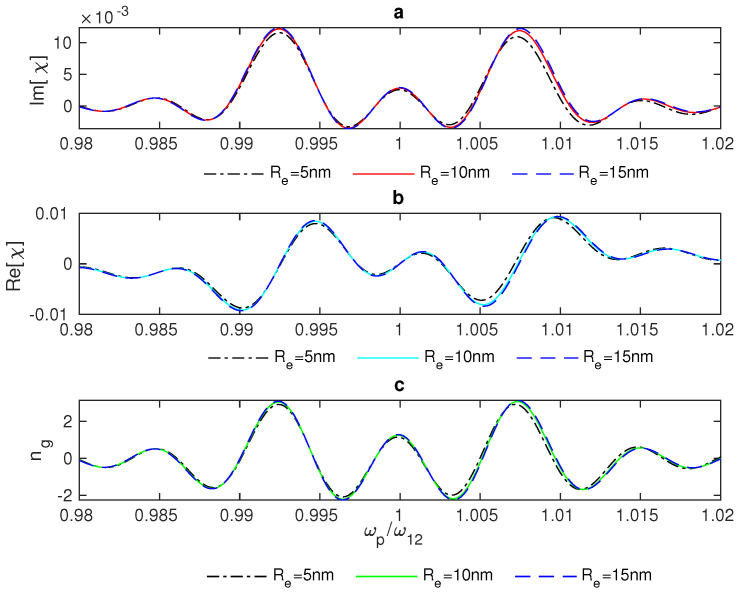
Linear optical properties of the hybrid system versus the probe field’s frequency: absorption (**a**), dispersion (**b**), and group index (**c**) of the system at Rabi frequency of the control field of $\Omega_{c}$ = 10 THz. The system interacts with a resonant control field ($\Delta_{c}$ = 0).

**Figure 10 nanomaterials-13-00834-f010:**
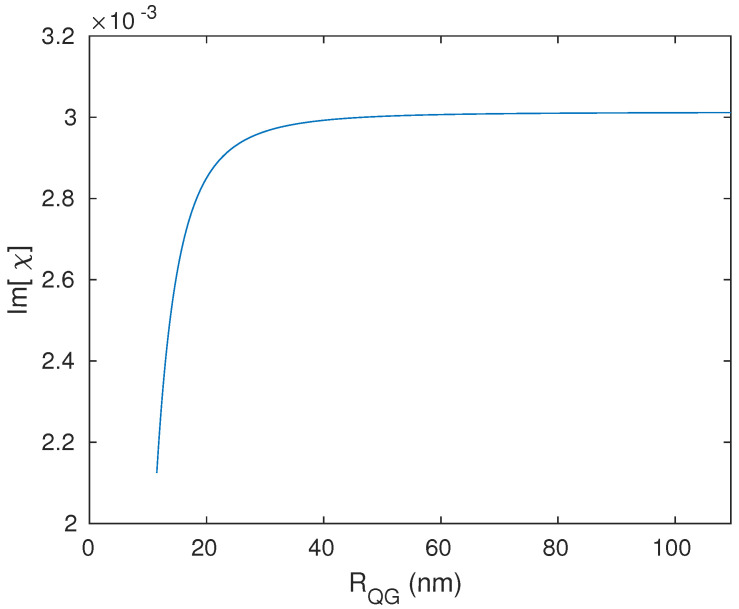
Resonant absorption of the hybrid system as a function of the distance $R_{QG}$ between the GND and QD with a Rabi frequency of $\Omega_{c}$ = 10 THz, and the system interacts with resonant external fields $\Delta_{p}$ = 0 and $\Delta_{c}$ = 0.

## Data Availability

Data is contained within the article.

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
