# Peer review of "Tunable Switching between Slow and Fast Light in the Graphene Nanodisks (GND)–Quantum Dot (QD) Plasmonic Hybrid Systems"

_nanomaterials, 2023, doi:10.3390/nano13050834_

Round 1
Reviewer 1 Report
In this paper, the authors reported the linear properties of the graphene nanodisks-quantum dots hybrid plasmonic systems in the near-infrared region of the electromagnetic spectrum by numerically solving the linear susceptibility of the weak probe field at a steady-state. It is an interesting phenomenon that the linear response of our hybrid plasmonic system exhibits an electromagnetically induced transparency window and switching between absorption and amplification without population inversion in the vicinity of the resonance. However, some minor problems should be solved in this paper before it is published in Nanomaterials. Below are my comments:
1. In the Introduction, appropriate deletions are recommended for the lengthy presentation of phthalic acid esters.
2. In this work, the linear optical response of hybrid system exhibits an EIT window and switching between absorption and amplification without population inversion near the resonance could be tuned. If possible, the authors should provide some explanations for this phenomenon.
3. Some formatting errors should be corrected, such the “nature” should be corrected to “Nature” in References 10.
Reviewer 2 Report
In this theoretical work, the authors intended to study the electromagnetically induced transparency (EIT) or absorption (EIA) using a hybrid system comprising a graphene nanodisk and a quantum dot with a three-level Lambda-type configuration. Their theoretical calculations are essentially the atomic model that has been extensively studied in atomic ensembles. By following the same derivations (such as rotating wave approximation, steady-state approximation, and dipole interaction), the authors further discussed the linear response associated with the considered EIT and EIA including slow and fast light effect, and transparency (or absorption) window width.
After carefully going through the manuscript, I found the work seems interesting. However, there are some severe issues appearing in the authors' analysis and discussions. In the following, I will specify the major issues for the authors to consider in the revision:
(1) I noticed that all the calculations are obtained by assuming the input EM fields are continuous waves, akin to the EIT and EIA implemented in the atomic systems. However, in their numerical simulations, the Rabi frequency of the control field is assumed to be at least at the THz level, which is too large to be impractical for a real cw laser. As such, all the simulations seem unphysical and cannot be realized in the lab. If the authors assume the fields to be pulsed lasers, then all the calculations cannot be implemented as the way shown in the current manuscript. Since many conclusions drawn in this work are based on the numerical simulations, I strongly suggest the authors to carefully check their model and numerical calculations for consistency, especially in terms of the reality.
(2) Since the system proposed here only contains one graphene nanodisk and one quantum dot, the expected EIT and EIA effects won't be easy to be achieved. Part of the reasons come from the insufficient interaction to induce the destructive and constructive interference for exciton transitions, as there is only one quantum dot. This is understandable if one recalls the collective enhancement factor N using atomic ensemble. Considering this drawback, I suggest the authors to tone down their claimed EIT and EIA effect in the proposed hybrid system.
(3) Minor issue: English can be improved and some grammar issues and typos can be corrected.
In short, I don't believe the current manuscript meets the high standards of nanomaterials. The main objection comes from the serious technical drawbacks existing in the presentation. I strongly suggest the authors to double check the mode and their numerical simulations in terms of the practical physical parameters. Otherwise, the reported results are misleading and won't be physically feasible in a lab.
Round 2
Reviewer 2 Report
I have carefully checked the revised manuscript as well as the response from the authors to my previous report.
For the laser power and the associated Rabi frequencies, it might be possible to use the high power CW fiber lasers to solve the issue. This is fine to me.
However, for the single quantum dot issue, I completely disagree with the authors' reply. I believe the authors messed up the composite atoms for one quantum dot with the net effect of one quantum dot. In other words, although a quantum dot is formed by hundreds and thousands of atoms, the net effect is only for "one" artificial atom. The quoted atomic density in the response as well as in the revision won't play the role of the atomic density in atomic ensembles. As such, this issue remains unresolved and I strongly suggest the authors to revise this part of content in the manuscript.
Proposing a phenomenon is fine. But exaggerating the possible effect is deleterious to the community.
